# Geochemical Characteristics and Their Geological Significance of Lower Cambrian Xiaoerblak Formation in Northwestern Tarim Basin, China

**Jianfeng Zheng [1,2,*], Yongjin Zhu [1,2,*], Lili Huang [1,2], Guo Yang [3] and Fangjie Hu [3]**

[1] PetroChina Hangzhou Research Institute of Geology, Hangzhou 310023, China; huangll_hz@petrochina.com.cn

[2] Key Laboratory of Carbonate Reservoirs CNPC, Hangzhou 310023, China

[3] PetroChina Tarim Oil Field Company, Korla 841000, China; yangguo-tlm@petrochina.com.cn (G.Y.); hufj-tlm@petrochina.com.cn (F.H.)

* Correspondence: zhengjf_hz@petrochina.com.cn (J.Z.); zhuyj_hz@petrochina.com.cn (Y.Z.)

**Abstract:** Lower Cambrian Xiaoerblak Formation is one of the major exploration targets in Cambrian pre-salt Tarim Basin; however, the exploration breakthrough is restricted by insufficient understanding of its sedimentary evolution and reservoir genesis. In this paper, based on a systematic description of the outcrop in the Xiaoerblak section, northwestern Tarim Basin, some samples were selected for tests of stable carbon and oxygen isotopic compositions, strontium isotopic composition, order degree, trace and rare earth elements, U-Pb isotopic age and clumped isotope. It is found that the Xiaoerblak Formation mainly develops nine types of dolomites, i.e., laminated microbial dolomite, thrombolite dolomite, stromatolite dolomite, foamy microbial dolomite, grain dolomite, etc. According to the lithofacies associations, it can be divided into three members: Xi 1, Xi 2, and Xi 3, of which member Xi 2 is subdivided into three submembers. The characteristics of lithofacies assemblage formed bottom to top indicate that it can be described as a third-order sequence. The Xiaoerblak Formation was deposited in a nearshore shallow seawater environment characterized by high water salinity and temperature under a warm and humid climate during the Early Cambrian, giving rise to the sedimentary sequence of inner ramp lagoon, subtidal microbial mound shoal and tidal flat in the carbonate ramp setting from bottom to top. Its dolomitization occurred in the penecontemporaneous–shallow burial period when the temperature was relatively low and high-salinity seawater acted as the main dolomitizaiton fluid. The reservoir space mainly comprises primary microbial framework pores and vugs formed by the atmospheric freshwater dissolution. Reservoirs were controlled by lithofacies, high-frequency sequence boundary and early dolomitization. The research results are of great significance for presalt Cambrian lithofacies paleogeographic mapping and reservoir prediction.

**Keywords:** dolomite; geochemical characteristics; sedimentary environment; pore genesis; lower Cambrian; Tarim Basin

## 1. Introduction

The presalt Cambrian in the Tarim Basin holds a huge quantity of oil and gas resources and has good reservoir-caprock assemblages, making it a strategic prospect for discovering large oil and gas fields [1–4]. It did not reveal any breakthrough in nearly 20-years of exploration until 2012 when Well ZS1 was successfully drilled, which suggested excellent reservoir-forming conditions in presalt Cambrian [5]. However, the successive failure of Wells YL6, XH1, CT1 and HT2 put the direction and potential of exploration in the system under question. In 2019, the Tarim Oilfield Company listed presalt Cambrian as one of the three major risk exploration targets. Consequently, the research on it was further strengthened. In 2020, Well LT1 obtained commercial oil and gas flows at 8200 m, recording a significant breakthrough in the ultra-deep presalt Cambrian strata in the Tarim Basin.





Thus, the prospectors became firmly confident and determined to look for large oil and gas fields in this basin.

The Lower Cambrian Xiaoerblak Formation is a major exploration target in the presalt Cambrian in the Tarim Basin and also a research hotspot in recent years. Many scholars have worked a lot to clarify the sedimentary facies. Wei, G.Q. reported that the sedimentary environment evolved from open-sea shelf to an evaporate platform in the Early Cambrian [6]. Ni, X.F. believed that the ramp platform was developed during the deposition of Xiaoerblak Formation, with the platform margin in the north [7]. Li, B.H. proposed that the platform margin was developed in the Keping area, where the grain shoal facies belt was most favorable [8]. Hu, M.Y. indicated that mixed tidal flat, a restricted–semi-evaporate platform, open platform, platform margin and slope1–basin facies were developed in the basin with the palaeogeographic pattern of higher west than east and higher south than north [9]. Zheng, J.F. demonstrated that the ramp-type carbonate platform characterized by a sedimentary system of microbial mat–mound beach–tidal flat was developed in the Xiaoerblak period [10]. Qiao, Z.F. proposed that microbial mounds and algal-psammitic shoals occurred in the Keping-Bachu area under the background of homogeneous tilted ramp [11]. Some scholars have investigated the reservoirs in the Xiaoerblak Formation. Li, Y. [12], Huang, Q.Y. [13], Deng, S.B. [14], and Yu, H.Y. [15] reported that microbialite is dominant and reservoir development was controlled by palaeogeomorphology, diagenetic processes and microbial structure. Shen, A.J. [16], Yan, W. [17], Wang, S. [18] and Zheng, J.F. [19] classified the reservoirs of microbial reef-beach facies as high-quality ones, and identified the sedimentary facies and atmospheric freshwater dissolution in the early supergene period as the main contributors to reservoir development. Li, B. [20] believed that the reservoirs were reworked by a variety of dolomitization processes (mainly buried dolomitization) and also by hydrothermal processes. Obviously, the understanding of sedimentary facies, dolomite genesis and reservoir genesis in the Xiaoerblak Formation are still unclear, which makes it difficult to predict favorable exploration zones of Xiaoerblak Formation.

Geochemical characteristics of rock minerals are indicative of sedimentary environment and diagenetic environment. In this paper, based on a systematic description of the outcrop in the Xiaoerblak section, northwestern Tarim Basin, some samples were selected for tests of stable carbon and oxygen isotopic compositions, strontium isotopic composition, trace elements and others, in order to investigate the paleosedimentary environment and dolomite forming environment during the deposition of the Xiaoerblak Formation. Moreover, the samples of carbonate cements in pores of dolomite reservoirs were analyzed in light of clumped isotopes and laser U-Pb isotopic age, aiming to clarify the pore-forming environment. The results are of great significance for research on lithofacies paleogeography and reservoir genesis/distribution of the Lower Cambrian Xiaoerblak Fromation in the Tarim Basin.

## 2. Geological Setting

The Tarim Basin in southern Xinjiang, China (Figure 1a), is a multi-cycle petroliferous basin superimposed by the Paleozoic cratonic basin and the Mesozoic and Cenozoic foreland basins [21], covering an area of $56 \times 10^4$ km². According to the current division of tectonic units, the basin is composed of seven first-order tectonic units, including Tabei Uplift, Central Uplift, Southeast Uplift, Kuqa Depression, North Depression, Southeast Depression and Southwest Depression (Figure 1b). The Xiaoerblak outcrop is found in the northwest margin of the basin, about 50 km southwest of Aksu, and is a part of the eastern segment of the Keping fault-uplift in the Tabei Uplift [22,23].

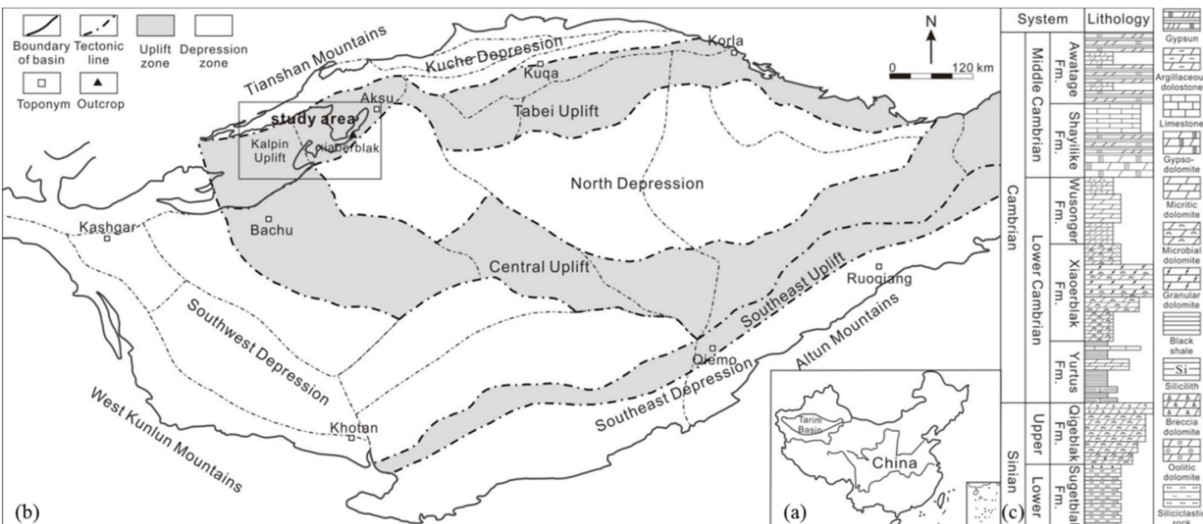

**Figure 1.** Geological setting map of study area: (**a**) Location of the Tarim Basin in northwestern China; (**b**) Tectonic location of the study area indicated with a grey rectangle; (**c**) Sinian-Lower Cambrian-Middle Cambrian stratigraphic framework.

The Precambrian tectonic evolution of the Tarim Basin was closely related to the amalgamation–breakup cycle of the Columbia supercontinent and the Rodinia super-continent [24]. As a result of the Rodinia supercontinent breaking up at the end of the Neoproterozoic, in the Nanhua–Early Sinian, continental extension stayed dominant, the Tarim plate evolved as a rift system, and the paleotectonic pattern characterized by one depression between two uplifts was formed. In the Late Sinian, the basin inherited the preceding paleotectonic pattern, and the continuous rifting led to the depression generated from the peripheral subsidence of the ocean basin, forming a depression basin dominated by a littoral–neritic carbonate platform [25,26]. At the end of Sinian, the Keping movement induced intensive uplifting within the Tarim plate, giving rise to widespread unconformity between Sinian and Cambrian strata. In the Early Cambrian, the Tarim Basin was in a ramp–shelf sedimentary system during the deposition of the Yuertus Formation, and formed high-quality source rocks after being transformed by rapid transgression [27]. Later, the whole basin stayed in a stage of slow regression, and the paleostructural pattern consisting of three uplifts and two depressions controlled the post-rift subsidence of the carbonate ramp system during the deposition of Xiaoerblak [28]. As the paleo-climate became hot and dry and the platform evolved with a barrel-shaped structure, thick evaporites were widely developed in the Middle Cambrian, forming the regional high-quality caprocks [29].

The Sinian–Cambrian strata are cropped out completely in the Xiaoerblak section, where the Lower Cambrian Yuertus Formation is in parallel unconformable contact with the Upper Sinian Qigbulak Formation, and the Lower Cambrian Xiaoerblak Formation is in conformable contact with the Yuertus Formation and the Wusonger Formation (Figure 1c). The Xiaoerblak Formation is measured to be about 158.3 m thick. According to the color, lithology, sedimentary microfacies and other features, it can be divided into three members: Xi 1, Xi 2 and Xi 3, of which member Xi 2 is subdivided into Xi $2^1$, Xi $2^2$ and Xi $2^3$ submembers (Figure 2). Member Xi 1 is composed of black-gray laminated microbial dolomite, with numerous flat dissolved pores/vugs filled with sparry calcite or saddle dolomite (LMD) (Figure 3a,b). Submember Xi $2^1$ mainly consists of dark gray layered thrombolite dolomite (LTD) (Figure 3c,d), and contains bedding-parallel flat dissolved pores/vugs filled or semi-filled with sparry calcite locally. Submember Xi $2^2$ is dominated by gray massive thrombolite dolomite (MTD) (Figure 3e,f), with bedding-parallel flat dissolved pores/vugs locally, where rare cements are observed. Submember Xi $2^3$ mainly consists of light gray–off-white thick–massive grain dolomite with bonding structure (GDBS) (Figure 3g), foamy microbial dolomite (FMD) (Figure 3h,i) and crystalline dolomite

remaining grain apparition (CDGA) (Figure 3j). Particularly, FMD contains a great number of pores, and CDGA is speculated to originate from grainstone [21]. Member Xi 3 is dominated by gray medium-layered stromatolite dolomite (SD), with laminated (LSD) and hummocky (HSD) structures (Figure 3k) and gray and yellowish gray laminated argillaceous dolomite (AD), grain dolomite with clay (GDC), micritic dolomite (MD) and yellow mudstone (Figure 3l), where a tepee structure and mud cracks are commonly observed. The Gamma Ray (GR) values are relatively high and greatly fluctuate in member Xi 3, and are generally low in other members with small variations.

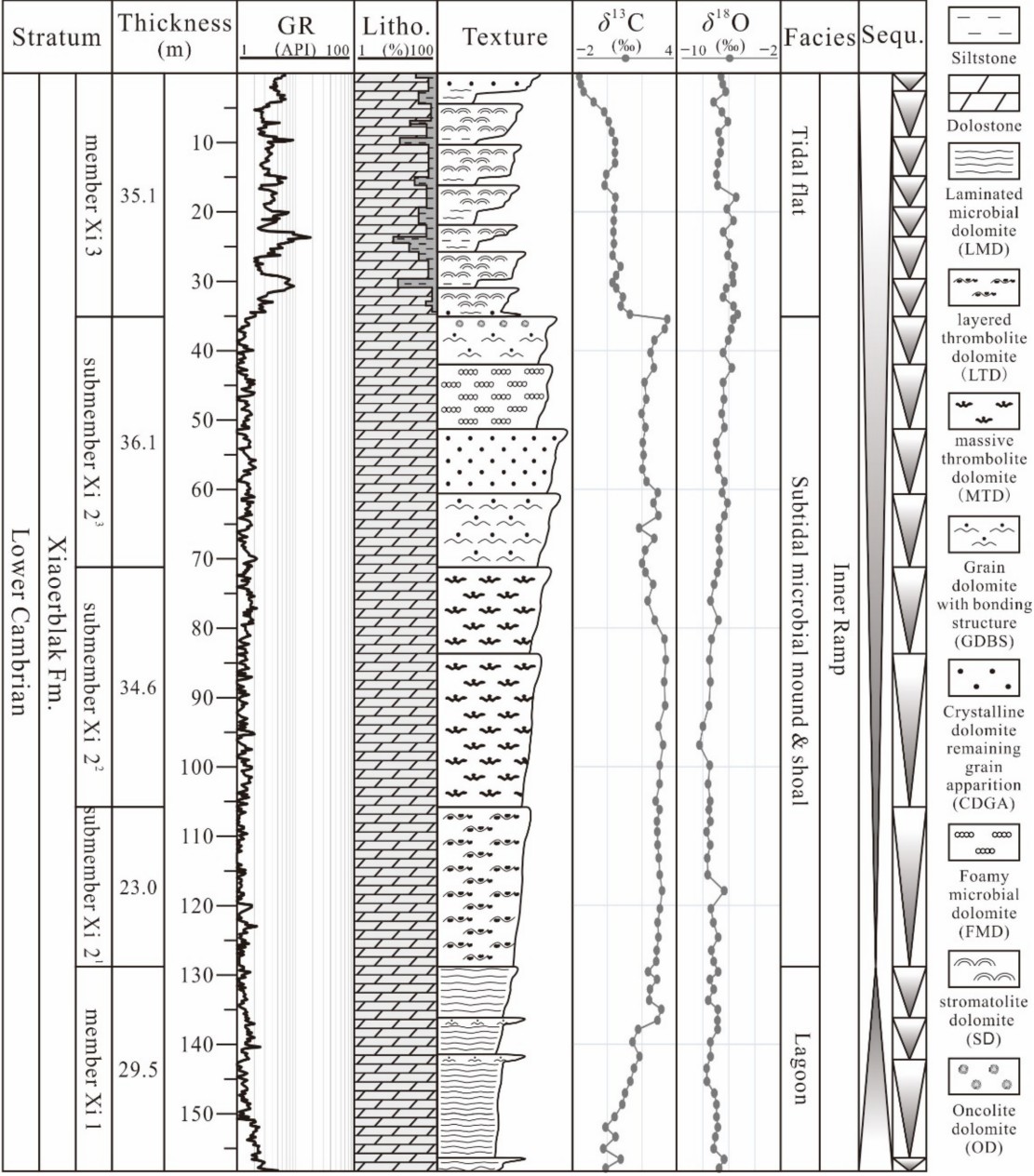

**Figure 2.** Stratigraphic column of Xiaoerblak Formation.

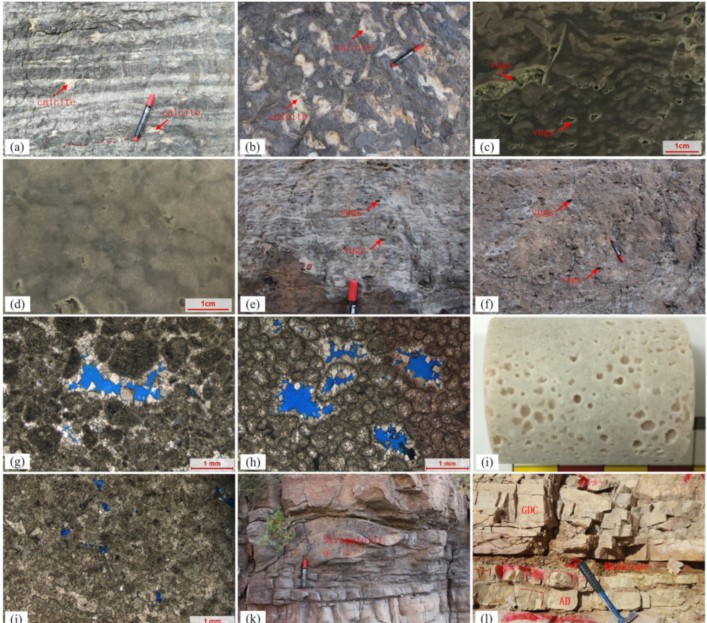

**Figure 3.** Photographs of dolomite in Xiaoerblak Formation: (**a**) laminated microbial dolomite, with alternating microwave bright and dark laminae, vugs filled with sparry calcite are developed along the bedding, member Xi 1, outcrop image; (**b**) Laminar dolomite (vertical perspective), member Xi 1, outcrop image; (**c**) Thrombolite dolomite, irregular layered structure, submember Xi $2^1$, sample section image; (**d**) Layered thrombolite dolomite, bedding vugs are uniformly distributed, submember Xi $2^1$, sample section image; (**e**) Thrombolite dolomite, reticular structure, submember Xi $2^2$, outcrop image; (**f**) Massive thrombolite dolomite, bedding vugs are uniformly distributed, submember Xi $2^2$, outcrop image; (**g**) Grain dolomite, with microbial bonding structure and a small amount of dissolved pores between grains, submember Xi $2^3$, blue casting thin section PPL image; (**h**) Foamy microbial dolomite, with fenestral dissolved pores, submember Xi $2^3$, blue casting thin section PPL image; (**i**) Foamy microbial dolomite, subround fenestra pores are uniformly distributed, submember Xi $2^3$, plug sample image; (**j**) Crystalline dolomite remaining grain apparition, with intercrystalline dissolved pores, submember Xi $2^3$, blue casting thin section PPL image; (**k**) Stromatolite dolomite, in low-amplitude moundy shape, member Xi 3, outcrop image; (**l**) Argillaceous dolomite, micritic dolomite and mudstone alternated with mud dolomite, member Xi 3, outcrop image.

## 3. Samples and Methods

A total of 110 fresh samples not weathered or altered were collected from the Xiaoerblak Formation in the Xiaoerblak section, the Tarim Basin. Casting thin sections were prepared for all samples. An analysis of stable carbon and oxygen isotopic compositions was conducted for 89 samples to establish the $\delta^{18}O$ and $\delta^{13}C$ trend line. Specifically, 18 of the 89 samples, covering the 9 types of dolomites in three members of the Xiaoerblak Formation, were analyzed for strontium isotopic composition, order degree, trace element and rare earth elements (REEs), respectively (Table 1). In order to avoid the interaction between carbonate cements and host rocks, a part with single-structure components was drilled from each of the samples using a micro-sampler. All drilled samples were grounded individually with an agate mortar to 200 mesh powder, and packaged separately with transparent drawing paper [30]. Moreover, for identifying the genesis of numerous dissolved pores filled or semi-filled with sparry calcite and saddle dolomite in the microbial dolomites in the middle–lower part of Xiaoerblak Formation, seven samples were selected for LA-ICP-MS U-Pb dating (five of the samples were developed dissolution pores cemented by carbonate minerals and two samples were sparry calcite from fractures), all the samples needed to be cut and cleaned before target preparation, then a 25mm diameter target was made (Figure 4). The method for preparing the target is similar to that of SHRIMP zircon targe [31]; in order to eliminate Pb contamination during sample preparation, it is necessary

to apply a super-clean treatment to the target in the super-clean chamber before testing. Five carbonate cements were selected for cluster isotope testing; the sample pretreatment method is the same as that for the carbon and oxygen isotope test samples.

**Table 1.** Order degree, $\delta^{13}$C, $\delta^{18}$O and $^{87}$Sr/$^{86}$Sr value of dolomite in Xiaoerblak formation.

| Sample | Lithofacies | Member | $\delta^{13}$C ‰ (PDB) | $\delta^{18}$O ‰ (PDB) | $^{87}$Sr/$^{86}$Sr $\pm 2\delta$ | Order Degree |
|---|---|---|---|---|---|---|
| 13–1 | laminated microbial dolomite (LMD) | Xi 1 | 0.88 | −6.93 | 0.709374 ± 3 | 0.67 |
| 14–1 | Laminated microbial dolomite (LMD) | Xi 1 | 1.34 | −7.67 | 0.708992 ± 8 | 0.50 |
| 20–2 | Laminated microbial dolomite (LMD) | Xi 1 | 2.43 | −7.56 | 0.709103 ± 5 | 0.54 |
| 25–1 | Layered thrombolite dolomite (LTD) | Xi 2$^1$ | 2.85 | −7.33 | 0.708843 ± 6 | 0.55 |
| 31–1 | Layered thrombolite dolomite (LTD) | Xi 2$^1$ | 2.91 | −7.42 | 0.709289 ± 13 | 0.59 |
| 35–2 | Massive thrombolite dolomite (MTD) | Xi 2$^2$ | 3.03 | −7.45 | 0.709167 ± 7 | 0.60 |
| 38–2 | Massive thrombolite dolomite (MTD) | Xi 2$^2$ | 3.35 | −7.51 | 0.709189 ± 4 | 0.68 |
| 43–5 | Massive thrombolite dolomite (MTD) | Xi 2$^2$ | 2.36 | −7.11 | 0.709122 ± 8 | 0.65 |
| 49–3 | Grain dolomite with bonding structure (GDBS) | Xi 2$^3$ | 1.86 | −6.32 | 0.709077 ± 7 | 0.62 |
| 51–1 | Crystalline dolomite remaining grain apparition (CDGA) | Xi 2$^3$ | 2.04 | −6.78 | 0.709113 ± 17 | 0.59 |
| 52–2 | Foamy microbial dolomite (FMD) | Xi 2$^3$ | 2.21 | −6.31 | 0.709105 ± 6 | 0.45 |
| 54–2 | Foamy microbial dolomite (FMD) | Xi 2$^3$ | 2.15 | −6.43 | 0.709113 ± 10 | 0.56 |
| 57–1 | Crystalline dolomite remaining grain apparition (CDGA) | Xi 2$^3$ | 2.73 | −6.02 | 0.709089 ± 8 | 0.61 |
| 64–1 | Grain dolomite with bonding structure (GDBS) | Xi 3 | 0.52 | −5.71 | 0.709108 ± 4 | 0.58 |
| 68–1 | Laminated stromatolite dolomite (LSD) | Xi 3 | 0.36 | −6.4 | 0.709412 ± 6 | 0.60 |
| 72–1 | Grain dolomite with clay (GDC) | Xi 3 | −0.13 | −6.83 | 0.713218 ± 2 | / |
| 76–1 | Laminated stromatolite dolomite (LSD) | Xi 3 | 0.47 | −6.58 | 0.709545 ± 5 | 0.61 |
| 81–1 | Argillaceous dolomite (AD) | Xi 3 | −0.38 | −7.14 | 0.713343 ± 11 | / |

Xi 2$^1$, Xi 2$^2$ and Xi 2$^3$ are three submembers of member Xi 2.

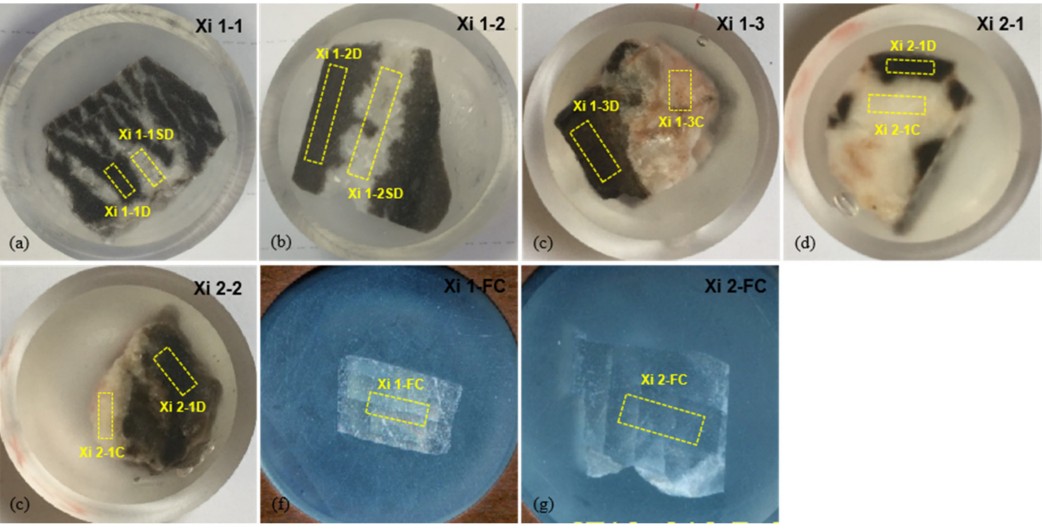

**Figure 4.** Photographs of sample target for U-Pb dating in Xiaoerblak Formation (the dotted boxes in the figure are the test points of different fabc): (**a**) Sample Xi 1-1; (**b**) Sample Xi 1-2; (**c**) Sample Xi 1-3; (**d**) Sample Xi 2-1; (**e**) Sample Xi 2-2; (**f**) Sample Xi 1-FC; (**g**) Sample Xi 2-FC.

Stable carbon, oxygen, strontium isotopic compositions, and order degree were tested at the CNPC (China National Petroleum Corporation, Hangzhou, China) Key Laboratory of Carbonate Reservoirs in Hangzhou. Stable carbon and oxygen isotopic compositions were measured using Delta V Advantage isotope ratio mass spectrometry (IRMS) on two reference samples, GBW4405 and GBW4406, with an accuracy of ±0.06‰ for $\delta^{13}$C and ±0.08‰ for $\delta^{18}$O. Strontium isotopic composition was measured with Triton Plus thermal ionization mass spectrometry (TIMS) on the reference sample GBW04411, with an accuracy higher than 0.01%. Order degree was measured using an X′pert Pro X-ray diffractometer, with a relative error <10%. Trace elements and REEs were tested at Tongwei Analytical Technology Co., Ltd., Guizhou, China, using the inductively coupled plasma mass spectrometer (ICP-MS), on two international reference samples, W-2a and BHVO-2, with a precision/accuracy greater than 5%. Laser ablation in-situ U-Pb dating of carbonate was performed at the Radiogenic Isotopes Laboratory, School of Geosciences, University of Queensland, Australia, using ASI Resolution SE, with a laser spot diameter of 100 μm, a frequency of 10 Hz and an energy density of 3 J/cm$^2$, together with a Nu Plasma II multi-collector inductively coupled plasma mass spectrometer (MC-ICP-MS), on two international reference samples, NIST 614 and 616. Based on the data processed with Lolite3.6, Isoplot3.0 was used to calculate the ages and map the Tera-Wasserburg concordia diagrams. Clumped isotope analysis was conducted at the Isotope Laboratory, University of Miami, using the Thermofisher Mat-253 gas stable isotope mass spectrometer, on four reference samples (ETH-1–ETH-4), with an accuracy of ±0.02‰.

## 4. Results

### 4.1. Stable Carbon and Oxygen Isotopic Compositions

Stable carbon and oxygen isotopic compositions are related to fluid medium causing dolomitization, and are mainly sensitive to the salinity and temperature of the medium. Thus, they can be used to identify the properties of dolomitization fluid and diagenetic environment. According to the cross-plot of $\delta^{13}$C and $\delta^{18}$O for 89 samples (Figure 5a), there is no obvious correlation between $\delta^{13}$C and $\delta^{18}$O, indicating that the samples were weakly transformed by late diagenesis and basically retained the basic information of their diagenetic fluids. The $\delta^{13}$C and $\delta^{18}$O values range from −1.21‰ to 3.48‰ and from −8.24‰ to −5.40‰, with averages of 1.70‰ and −6.77‰, respectively. The $\delta^{13}$C value differs among members: it varies from −1.21‰ to 1.32‰ (avg. 0.21‰), generally in negative range, in member Xi 3, from −0.08‰ to 3.12‰ (avg. 1.43‰) in member Xi 1, and from 1.79‰ to 3.48‰ (avg. 2.64‰) in member Xi 2 with just a slight difference among three submembers. Moreover, the $\delta^{18}$O values fall in the range (from −6‰ to −8‰) derived by Veizer on the basis of the Early–Middle Cambrian global seawater [30]. Particularly, the $\delta^{18}$O values in Submember Xi 2$^3$ and member Xi 3 (avg. −6.30‰) in the mid-upper part are slightly more positive than those in the strata in the middle-lower part (avg. −7.25‰).

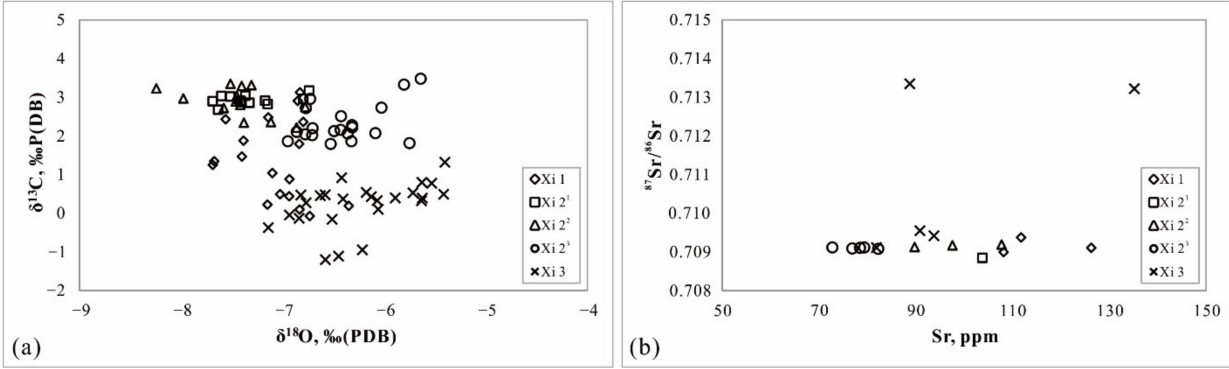

**Figure 5.** Binary plots of Geochemistry in Xiaoerblak Formation dolomite: (**a**) $\delta^{13}$C versus $\delta^{18}$O; (**b**) $^{87}$Sr/$^{86}$Sr versus Sr content.

### 4.2. Strontium Isotopic Composition

Strontium isotopic composition in carbonate rocks is affected by environment and fluid, and is one of the important indicators of paleoclimate and diagenetic fluid. As shown in Table 1, the $^{87}Sr/^{86}Sr$ value varies between 0.708843 and 0.713343, with an average of 0.709629. The $^{87}Sr/^{86}Sr$ values of two clay-bearing samples from member Xi 3 were measured as 0.713218 and 0.713343, respectively, which are significantly higher than that of samples without clay. Excluding the two clay-bearing samples, the $^{87}Sr/^{86}Sr$ values are consistent in members and types of dolomites (Figure 5b), with an average of 0.709165, which is generally close to the average value of $^{87}Sr/^{86}Sr$ (0.7090) derived by Denison on the basis of the Early–Middle Cambrian global seawater [32].

### 4.3. Order Degree of Dolomite

Order degree is an important measure of crystallization rate, crystallization temperature and evolution degree of dolomite [33]. According to the test results (Table 1), for 18 samples, excluding two clay-bearing samples from which no data were measured, the order degree is generally 0.45–0.67 (avg. 0.59), suggesting a low range.

### 4.4. Trace Elements and REEs

Element geochemistry of carbonate rocks can illustrate sedimentary and diagenetic conditions, such as paleoclimate, redox environment, and water depth [34]. As shown in Table 2, all samples exhibit an MgO/CaO value close to 0.71, varying from 0.694 to 0.722. Except for two clay-bearing dolomite samples (72–1 and 81–1) with relative low MgO + CaO value (40.05–31.64%), the remaining samples have an MgO + CaO value from 50.75% to 52.08%, indicating a high purity. The contents of Al, Fe and Na are relatively high and vary greatly—Al: 100–9090 ppm, avg. 1079 ppm; Fe: 210–15740 ppm, avg. 2428 ppm; Na: 240–1050 ppm, avg. 527 ppm. The contents of Sr and Mn are medium and vary slightly—Sr: 72.8–159.1 ppm, avg. 99.2 ppm; Mn: 90–373 ppm, avg. 196 ppm. The contents of V, Cr, Ni, Cu, Ga, Ba and Pb are low, with average values of less than 10 ppm. From bottom to top, the contents of Sr, Na, Ba, Ni, Mn, Cr and V are relatively stable or slightly increase, and the contents of Al, Fe, Pb, Ga, Zn and Cu tend to decrease slowly and then increase rapidly (Figure 6).

**Table 2.** Trace elements content of dolomite in Xiaoerblak Formation.

| Sample | V (ppm) | Cr (ppm) | Mn (ppm) | Ni (ppm) | Cu (ppm) | Zn (ppm) | Ga (ppm) | Sr (ppm) | Ba (ppm) | Pb (ppm) | K (%) | Na (%) | Fe (%) | Al (%) | CaO (%) | MgO (%) |
|---|---|---|---|---|---|---|---|---|---|---|---|---|---|---|---|---|
| 13–1 | 6.86 | 1.19 | 180 | 6.19 | 2.69 | 3.74 | 0.12 | 111.8 | 3.68 | 0.54 | <0.01 | 0.04 | 0.028 | 0.038 | 29.96 | 21.63 |
| 14–1 | 7.77 | 4.43 | 283 | 7.08 | 2.55 | 14.1 | 0.1 | 108.2 | 3.41 | 0.89 | 0.01 | 0.041 | 0.134 | 0.025 | 30.01 | 21.48 |
| 20–2 | 7.1 | 3.51 | 245 | 6.62 | 2.66 | 8.95 | 0.11 | 126.3 | 4.92 | 0.77 | <0.01 | 0.048 | 0.109 | 0.028 | 30.04 | 21.37 |
| 25–1 | 6.67 | 4.9 | 271 | 6.6 | 2.73 | 9.01 | 0.1 | 159.1 | 7.67 | 0.89 | <0.01 | 0.063 | 0.164 | 0.022 | 30.06 | 21.61 |
| 31–1 | 6.26 | 4.2 | 322 | 6.81 | 3.49 | 9.61 | 0.12 | 103.8 | 7.98 | 2.08 | <0.01 | 0.039 | 0.23 | 0.03 | 29.82 | 21.29 |
| 35–2 | 5.75 | 3.26 | 239 | 7.05 | 2.92 | 8.25 | 0.08 | 97.6 | 5.85 | 1.25 | <0.01 | 0.048 | 0.155 | 0.023 | 29.94 | 21.57 |
| 38–2 | 5.23 | 1.81 | 205 | 6.04 | 2.25 | 9.33 | 0.07 | 107.8 | 5.74 | 1.04 | <0.01 | 0.051 | 0.122 | 0.02 | 30.19 | 21.61 |
| 43–5 | 5.02 | 2.12 | 157 | 6.56 | 3.21 | 5.05 | 0.05 | 89.8 | 4.04 | 1.01 | <0.01 | 0.038 | 0.106 | 0.014 | 29.69 | 21.06 |
| 49–3 | 5.24 | 1.31 | 128 | 7.45 | 2.11 | 4.63 | 0.04 | 82.3 | 14.2 | 0.78 | <0.01 | 0.049 | 0.067 | 0.015 | 30.32 | 21.18 |
| 51–1 | 5.17 | 1.26 | 141 | 6.42 | 1.87 | 6.21 | 0.04 | 72.8 | 11.3 | 0.39 | <0.01 | 0.097 | 0.058 | 0.013 | 30.56 | 21.52 |
| 52–2 | 5.47 | 0.45 | 90 | 9.33 | 1.01 | 2.66 | 0.02 | 78.5 | 27.3 | 0.84 | <0.01 | 0.024 | 0.021 | 0.01 | 30.19 | 21.7 |
| 54–2 | 5.28 | 1.37 | 129 | 7.49 | 2.33 | 4.64 | 0.04 | 79.4 | 13.53 | 0.91 | <0.01 | 0.053 | 0.062 | 0.012 | 30.15 | 21.76 |
| 57–1 | 5.33 | 1.12 | 127 | 7.67 | 1.83 | 4.55 | 0.04 | 76.9 | 16.89 | 0.74 | <0.01 | 0.056 | 0.052 | 0.014 | 30.45 | 21.61 |
| 64–1 | 8.21 | 3.53 | 176 | 8.81 | 3.52 | 18.41 | 0.18 | 81.9 | 14.35 | 1.61 | 0.04 | 0.042 | 0.265 | 0.021 | 29.77 | 21.12 |
| 68–1 | 9.64 | 3.22 | 150 | 9.95 | 3.46 | 26.7 | 0.2 | 93.8 | 12.2 | 1.38 | 0.03 | 0.031 | 0.119 | 0.077 | 30.44 | 21.43 |
| 72–1 | 15.62 | 11.1 | 373 | 17.83 | 29.3 | 95.9 | 1.33 | 135.2 | 23.2 | 20.7 | 0.23 | 0.105 | 0.925 | 0.569 | 23.66 | 16.42 |
| 76–1 | 8.39 | 3.31 | 164 | 9.78 | 5.27 | 23.28 | 0.25 | 90.9 | 15.48 | 3.24 | 0.06 | 0.056 | 0.179 | 0.102 | 29.69 | 21.24 |
| 81–1 | 15.66 | 7.89 | 148 | 18.8 | 9.89 | 68.3 | 1.12 | 88.8 | 28.1 | 19.1 | 0.2 | 0.068 | 1.574 | 0.909 | 18.49 | 13.15 |

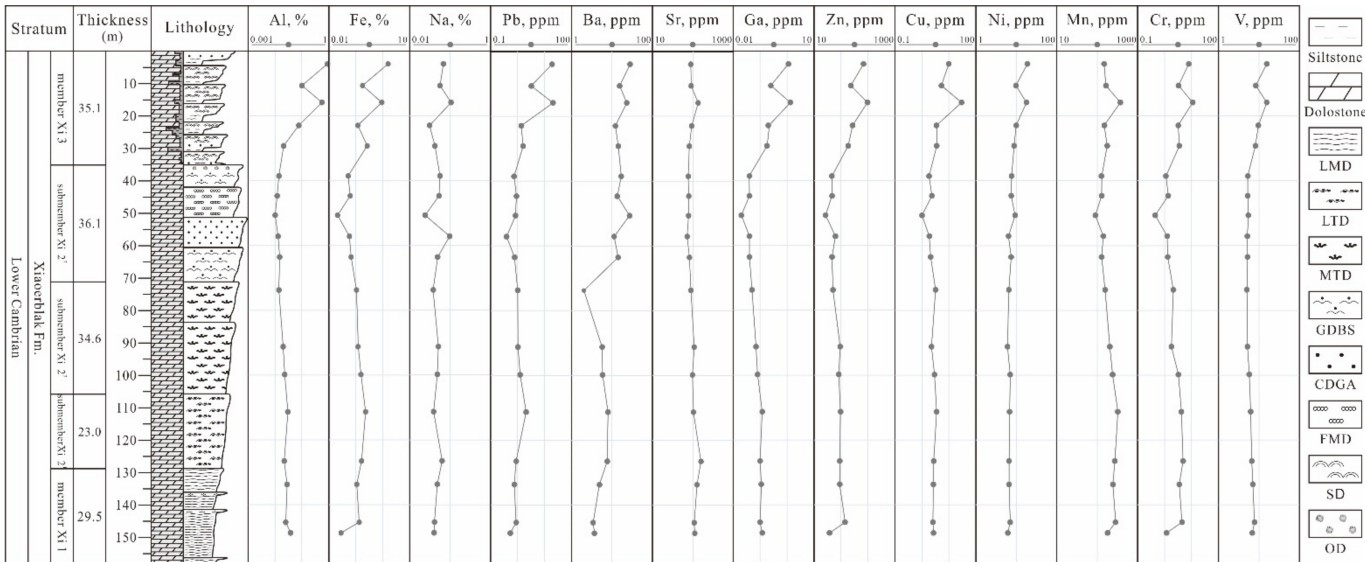

**Figure 6.** Geochemical variation tendency of Trace elements in Xiaoerblak Formation dolostone.

Rare earth elements (REEs) in carbonate minerals, with their relative abundance mainly dependent on their content and geochemical properties in dolomitization fluid, are weakly affected in the diagenetic process; this can effectively indicate the sedimentary environment or the origin of the dolomitization fluid [35]. As shown in Table 3, two clay-bearing dolomite samples (72–1 and 81–1) exhibit the highest ΣREE, being 38.16 ppm and 38.97 ppm, respectively, followed by the samples from member Xi 1 (avg. 8.72 ppm); the dolomite samples without clay from member Xi 2 and member Xi 3 have relatively low ΣREE, with an average of 1.26 ppm and 2.57 ppm, respectively. From bottom to top, ΣREE presents a trend of "high–low–high". The measured values were normalized to the Post-Archean Australian Shale (PAAS), and the normalized REEs anomalies were calculated by the following equations: $\delta Ce = 2 \times Ce_{SN}/(La_{SN} + Pr_{SN})$, $\delta Eu = 2 \times Eu_{SN}/(Sm_{SN} + Gd_{SN})$ [36]. The calculated $\delta Ce$ and $\delta Eu$ larger than 1.2 indicate a positive anomaly, and less than 0.8 indicate a negative anomaly. As illustrated by the PAAS-normalized REEs distribution patterns (Figure 7), member Xi 1 shows a wave pattern and member Xi 3 shows a hat pattern, while Xi 2 presents a wave pattern at the bottom, similar to member Xi 1, an up-dipping pattern in the middle part with $\delta Eu$ anomalies, and a hat pattern at the top, similar to member Xi 3.

### 4.5. U-Pb Dating

Laser ablation in-situ U-Pb dating of carbonate is an effective technique for determining the age of host rocks and cements of ancient carbonate rocks [37]. As shown in Table 4 and Figure 8, the ages of all carbonate cements were younger than those of host rocks; it shows that the test results conform to the sequence of diagenesis and are relatively reliable. The ages of the host rocks of the five microbial dolomites range from 480 ± 25 Ma to 501.7 ± 9.3 Ma, corresponding to the strata date from Furongian to Miaolingian of Cambrian. The ages of two saddle dolomites filled in the fractures are 14 ± 13 Ma and 45 ± 69 Ma respectively, corresponding to the strata dates of Neogence Miocene and Paleogene Eocene. The ages of three sparry calcites filled in the vugs range from 467 ± 17 Ma to 479 ± 16 Ma, corresponding to the strata dating from the Middle Ordovician to Lower Ordovician. The ages of two sparry calcites filled in the fractures are 3.18 ± 17 Ma and 209.8 ± 2.7 Ma, corresponding to the strata dates of Neogene Pliocene and Upper Triassic.

**Table 3.** REEs content of dolomite in Xiaoerblak Formation.

| Sample | La (ppm) | Ce (ppm) | Pr (ppm) | Nd (ppm) | Sm (ppm) | Eu (ppm) | Gd (ppm) | Tb (ppm) | Dy (ppm) | Ho (ppm) | Er (ppm) | Tm (ppm) | Yb (ppm) | Lu (ppm) | ΣREE (ppm) |
|---|---|---|---|---|---|---|---|---|---|---|---|---|---|---|---|
| 13–1 | 3.291 | 5.396 | 0.580 | 2.526 | 0.442 | 0.085 | 0.421 | 0.055 | 0.319 | 0.072 | 0.180 | 0.024 | 0.131 | 0.020 | 13.543 |
| 14–1 | 1.553 | 3.201 | 0.360 | 1.601 | 0.290 | 0.061 | 0.269 | 0.034 | 0.189 | 0.043 | 0.103 | 0.013 | 0.068 | 0.011 | 7.797 |
| 20–2 | 1.024 | 1.922 | 0.229 | 0.972 | 0.187 | 0.039 | 0.163 | 0.023 | 0.123 | 0.026 | 0.063 | 0.009 | 0.049 | 0.008 | 4.837 |
| 25–1 | 0.769 | 1.212 | 0.160 | 0.696 | 0.140 | 0.029 | 0.120 | 0.019 | 0.101 | 0.022 | 0.054 | 0.007 | 0.048 | 0.007 | 3.384 |
| 31–1 | 0.440 | 0.750 | 0.075 | 0.318 | 0.063 | 0.015 | 0.062 | 0.008 | 0.054 | 0.013 | 0.031 | 0.005 | 0.026 | 0.004 | 1.863 |
| 35–2 | 0.428 | 0.735 | 0.087 | 0.375 | 0.072 | 0.018 | 0.071 | 0.011 | 0.061 | 0.015 | 0.034 | 0.007 | 0.030 | 0.006 | 1.949 |
| 38–2 | 0.374 | 0.681 | 0.085 | 0.368 | 0.076 | 0.021 | 0.082 | 0.013 | 0.066 | 0.019 | 0.041 | 0.010 | 0.031 | 0.008 | 1.875 |
| 43–5 | 0.136 | 0.299 | 0.028 | 0.120 | 0.025 | 0.007 | 0.019 | 0.004 | 0.024 | 0.007 | 0.012 | 0.004 | 0.014 | 0.004 | 0.703 |
| 49–3 | 0.132 | 0.294 | 0.029 | 0.125 | 0.026 | 0.008 | 0.024 | 0.006 | 0.027 | 0.008 | 0.016 | 0.005 | 0.017 | 0.005 | 0.722 |
| 51–1 | 0.139 | 0.297 | 0.030 | 0.129 | 0.022 | 0.006 | 0.028 | 0.004 | 0.024 | 0.008 | 0.014 | 0.003 | 0.015 | 0.003 | 0.722 |
| 52–2 | 0.128 | 0.280 | 0.027 | 0.120 | 0.025 | 0.009 | 0.020 | 0.006 | 0.021 | 0.006 | 0.012 | 0.004 | 0.019 | 0.005 | 0.682 |
| 54–2 | 0.133 | 0.295 | 0.028 | 0.122 | 0.024 | 0.008 | 0.029 | 0.006 | 0.026 | 0.008 | 0.016 | 0.005 | 0.017 | 0.004 | 0.721 |
| 57–1 | 0.367 | 0.783 | 0.106 | 0.481 | 0.087 | 0.018 | 0.084 | 0.012 | 0.072 | 0.015 | 0.038 | 0.006 | 0.034 | 0.005 | 2.107 |
| 64–1 | 0.439 | 0.866 | 0.113 | 0.494 | 0.086 | 0.019 | 0.081 | 0.014 | 0.076 | 0.017 | 0.040 | 0.006 | 0.041 | 0.006 | 2.298 |
| 68–1 | 0.511 | 0.949 | 0.120 | 0.507 | 0.085 | 0.020 | 0.077 | 0.015 | 0.082 | 0.018 | 0.040 | 0.006 | 0.046 | 0.006 | 2.482 |
| 72–1 | 7.279 | 15.491 | 1.830 | 7.746 | 1.520 | 0.261 | 1.290 | 0.201 | 1.090 | 0.221 | 0.590 | 0.083 | 0.490 | 0.067 | 38.159 |
| 76–1 | 0.537 | 1.099 | 0.146 | 0.650 | 0.115 | 0.025 | 0.110 | 0.017 | 0.098 | 0.021 | 0.052 | 0.008 | 0.050 | 0.007 | 2.936 |
| 81–1 | 5.727 | 14.398 | 1.941 | 8.928 | 2.119 | 0.402 | 1.740 | 0.271 | 1.491 | 0.291 | 0.760 | 0.112 | 0.690 | 0.098 | 38.967 |

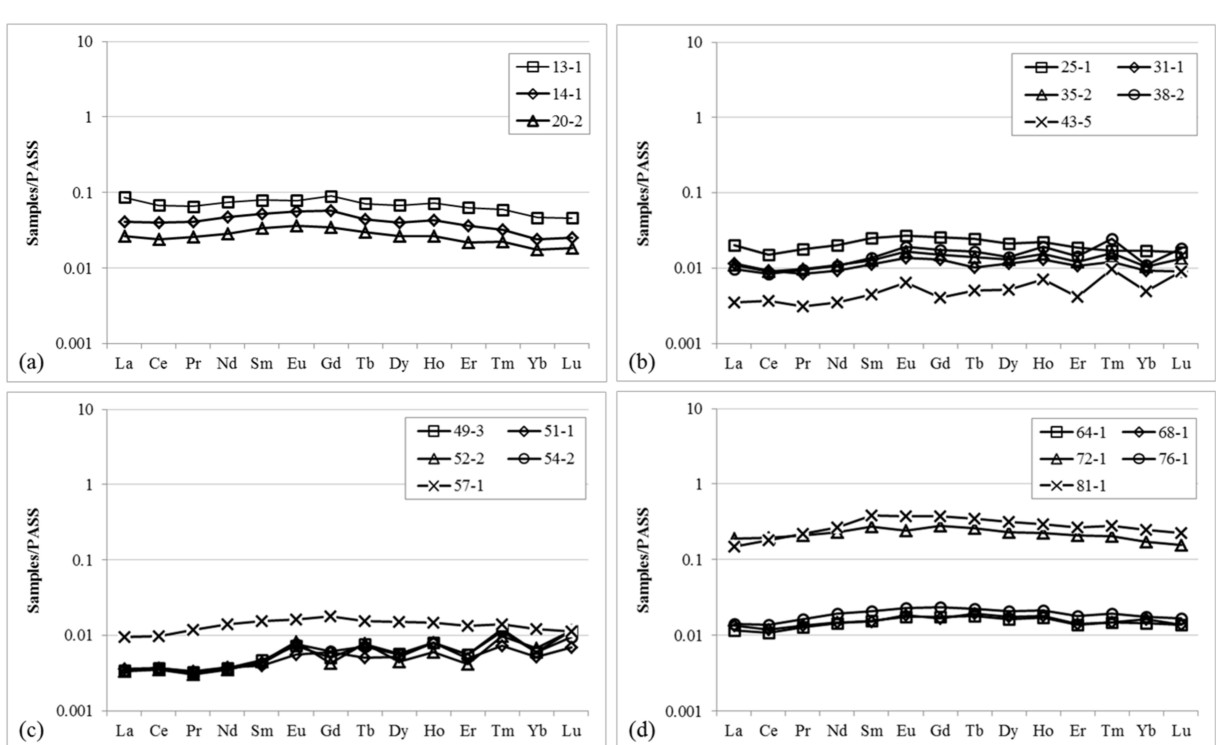

**Figure 7.** Normalized REEs distribution patterns of dolomite in Xiaoerblak Formation dolomite: (**a**) member Xi 1; (**b**) submember Xi $2^1$ and Xi $2^2$; (**c**) submember Xi $2^3$; (**d**) member Xi 3.

### 4.6. Clumped Isotope

Clumped isotope thermometry is an innovative experimental technique for determining the diagenetic temperature of carbonate minerals [38,39]. As shown in Table 5 and Figure 8, the carbonate cements of different occurrences were formed in different temperatures. The saddle dolomites filling in two dissolved fractures were formed at higher temperatures, 85.3 °C and 83.4 °C, and the $\delta^{18}O_w$ values of their diagenetic fluids were 1.938‰ and 1.33‰, respectively. The sparry calcite filling in the fractures was formed at

74 °C, and the $\delta^{18}O_w$ value of its diagenetic fluid was −2.82‰. The formation temperatures of sparry calcite filling in three dissolved pores/vugs are low (41.8 °C, 45.3 °C and 34 °C), and the $\delta^{18}O_w$ (SMOW) values of their diagenetic fluids were −5.00‰, −5.02‰ and −5.38‰, respectively.

**Table 4.** The result of U-Pb isotopic dating.

| Sample | Components | U-Pb Age Ma | Sample | Occurrence | U-Pb Age Ma |
|---|---|---|---|---|---|
| Xi 11 | Xi 11D (microbial dolomite) | 498 ± 25 | Xi 21 | Xi 21D (microbial dolomite) | 480 ± 25 |
| | Xi 11SD (Saddle dolomite in fracture) | 14 ± 13 | | Xi 21C (Sparry calcite in vug) | 467 ± 17 |
| Xi 1-2 | Xi 12D (microbial dolomite) | 501.7 ± 9.3 | Xi 22 | Xi 22D (microbial dolomite) | 495.6 ± 6.3 |
| | Xi 12SD (Saddle dolomite in fracture) | 45 ± 69 | | Xi 22C (Sparry calcite in vug) | 467 ± 12 |
| Xi 13 | Xi 13D (microbial dolomite) | 500 ± 25 | Xi 1FC | Sparry calcite in fracture | 209.8 ± 2.7 |
| | Xi 13C (Sparry calcite in vug) | 479 ± 16 | Xi 2FC | Sparry calcite in fracture | 3.18 ± 17 |

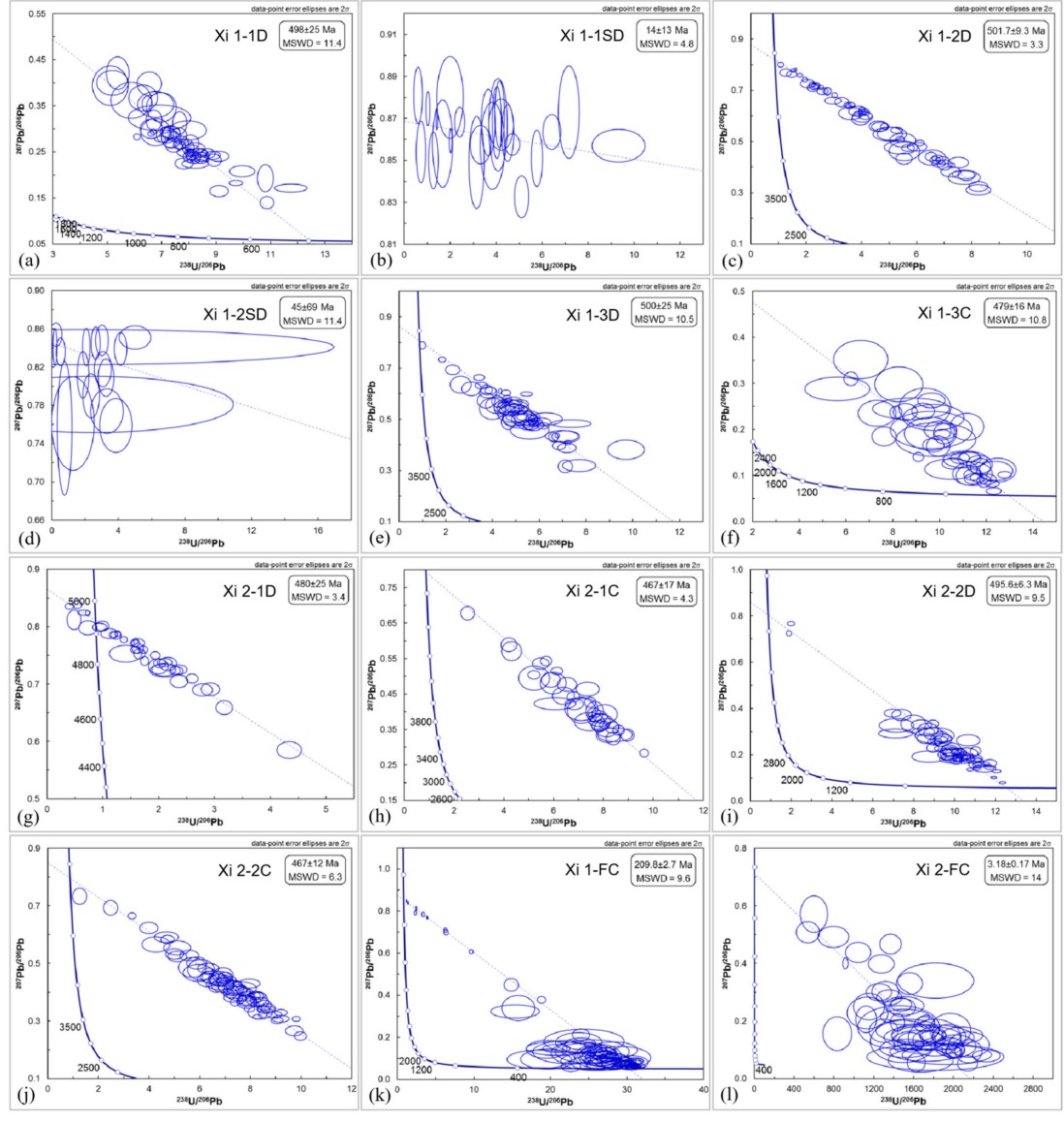

**Figure 8.** LA-MC-ICP-MS U-Pb concordia diagram of dolomite in Xiaoerblak Formation carbonate: (**a**) Microbial dolomite of sample Xi 1-1; (**b**) Saddle dolomite of sample Xi 1-1; (**c**) Microbial dolomite

of sample Xi 1-2; (**d**) Saddle dolomite of sample Xi 1-2; (**e**) Microbial dolomite of sample Xi 1-3; (**f**) Sparry calcite of sample Xi 1-3; (**g**) Microbial dolomite of sample Xi 2-1; (**h**) Sparry calcite of sample Xi 2-1; (**i**) Microbial dolomite of sample Xi 2-2; (**j**) Sparry calcite of sample Xi 2-2; (**k**) Sparry calcite of sample Xi 1-FC; (**l**) Sparry calcite of sample Xi 2-FC.

**Table 5.** The result of U-Pb isotopic dating.

| Sample | Components | $\delta^{13}Cm$ % (PDB) | Standard Deviation | $\delta^{18}Om$ % (PDB) | Standard Deviation | Δ47 | T °C | $\delta^{18}Ow$ % (SMOW) |
|---|---|---|---|---|---|---|---|---|
| Xi 1-1 | Xi 1-1SD (Saddle dolomite in fracture) | 0.87 | 0.01 | −10.59 | 0.01 | 0.555 | 85.3 | 1.93 |
| Xi 1-2 | Xi 1-2SD (Saddle dolomite in fracture) | −0.46 | 0.02 | −10.92 | 0.02 | 0.5559 | 83.4 | 1.33 |
| Xi 1-3 | Xi 1-3C (Sparry calcite in vug) | −2.49 | 0.01 | −10.57 | 0.01 | 0.647 | 41.8 | −5.00 |
| Xi 1-F | Xi 1-FC (Sparry calcite in fracture) | −0.68 | 0.02 | −13.66 | 0.03 | 0.576 | 74.0 | −2.82 |
| Xi 2-1 | Xi 2-1C (Sparry calcite in vug) | −2.25 | 0.05 | −9.5 | 0.05 | 0.668 | 34.0 | −5.38 |
| Xi 2-2 | Xi 2-2C (Sparry calcite in vug) | −2.81 | 0.01 | −11.21 | 0.01 | 0.638 | 45.3 | −5.02 |

## 5. Discussion

### 5.1. Sedimentary Environment

According to the variation of elements and isotopes of sedimentary rocks in sedimentary–diagenetic processes, the factors controlling the sedimentary environment, such as paleotemperature, paleosalinity, paleowater depth, redox conditions and paleoclimate, can be analyzed.

Usually, $\delta^{13}C$ and $\delta^{18}O$ increase with the salinity of fluid. $\delta^{13}C$ shows the closest relation to paleosalinity and is less affected by temperature. Keith et al. [40] established a formula for calculating the paleosalinity (Z) of seawater as follows: Z = 2.048 × ($\delta^{13}C$ + 50) + 0.498 × $\delta^{18}O$ + 50), where $\delta^{13}C$ and $\delta^{18}O$ are relative values to the PDB standard, and indicated that Z < 120 represents a freshwater environment and Z > 120 represents a seawater environment. Accordingly, the Z value of Xiaoerblak Formation dolomites is calculated to be 121.6–131.6 (avg. 127.4), indicating a high-salinity seawater environment favorable for microbial reproduction. From bottom to top, member Xi 1 shows a gradually increasing Z value, member Xi 2 keeps a relatively stable Z value, and member Xi 3 exhibits a decreasing Z value, suggesting that member Xi 1 and member Xi 3 were deposited in the near-shore shallow water environment under the significant control of atmospheric freshwater.

Water temperature is an important controller of stable carbon and oxygen isotopic compositions in carbonates; thus, the $\delta^{18}O$ value can be used as a reliable indicator for measuring paleotemperature [41]. Urey (1948) first proposed the method to determine the paleoocean temperature with $\delta^{18}O$. This method was embodied by Epstein (1951) and then modified by Craig (1965) to a specific calculation formula: T (°C) = 15.976 − 4.2 × $\delta^{18}O$ + 0.13 × ($\delta^{18}O$ + 0.22)$^2$. Considering the age effect, that is, the effect of diagenesis and deviation of $\delta^{18}O$ increase with age, the formula may generate some errors in the case of ancient carbonate samples. To avoid the age effect, Keith (1964) put forward a correction by $\delta^{18}O = \delta^{18}O$ measured-$\triangle\delta^{18}O$, where $\triangle\delta^{18}O$ is the difference between the average value of $\delta^{18}O$ for the target formation (−6.77‰ for the Xiaoerblak Formation in this study) and the average value of $\delta^{18}O$ for the Quaternary marine carbonate rocks (−1.2‰) [42]. Accordingly, the paleo-temperature for the Xiaoerblak Formation is calculated to be 15.3 °C–28.2 °C (avg. 21.3 °C), a temperature suitable for microbial reproduction.

The accumulation and dispersion of elements are related to water depth (distance from shore), as the result of mechanical differentiation, chemical differentiation and biochemical action in depositional processes [43]. Therefore, trace elements such as Al, Fe and Pb can effectively indicate the paleo-water depth. The contents of Mn (avg. 196 ppm) and Cr (avg. 3.4 ppm) are not high and show minor variations bottom up, suggesting that the Xiaoerblak Formation was generally deposited in a near-shore shallow water environment and has the characteristics of carbonate inner ramp facies. The contents of Fe, Pb, Ga, Zn and Cu increase significantly from bottom to top, indicating that the depositional water

varies from member to member. Specifically, member Xi 1 and member Xi 2 were deposited in relatively deep water, while member Xi 3 was deposited in a tidal flat environment with the smallest depth and distance to shore. In addition, the significant increase in Al and K contents in member Xi 3 indicate an influence of terrigenous materials, reflecting a tidal flat environment. Sr/Ba is an indicator of the restriction degree of water. A lager Sr/Ba value suggests a higher restriction of water. According to the calculation results, member Xi 1 has the highest Sr/Ba values (avg. 29.30), followed by submember Xi $2^1$ and submember Xi $2^2$ (avg. 18.3); submember Xi $2^3$ and member Xi 3 have the smallest Sr/Ba values, with an average of 5.4. Thus, it is inferred that member Xi 1 was deposited in a restricted lagoon environment.

V/(V + Ni) is a sensitive indicator of redox conditions in an aqueous medium. Generally, V/(V + Ni) < 0.45 reflects an aerobic sedimentary environment, V/(V + Ni) of 0.45–0.6 reflects an oxygen-deficient environment, and V/(V + Ni) > 0.6 reflects an anaerobic reducing environment [30]. The calculated V/(V + Ni) ranges from 0.37 to 0.53, with an average of 0.46, indicating that the oxygen content varied throughout the depositional period of the Xiaoerblak Formation. member Xi 1 and submember Xi $2^1$, with an average V/(V + Ni) of 0.52 and 0.49, respectively, are considered to have deposited in an oxygen-deficient reducing environment, indicative of restricted lagoon or low-energy subtidal deposition. submember Xi $2^2$, with an average V/(V + Ni) of 0.45, is believed to have deposited in an intermediate environment between oxygen-deficient and oxygen-enriched, indicative of medium- to low-energy subtidal deposition. Submember Xi $2^3$, with an average V/(V + Ni) of 0.41, is considered a relatively oxygen-enriched oxidation environment, indicative of medium- to high-energy intertidal–subtidal deposition. member Xi 1, with an average V/(V + Ni) of 0.47, considered to be an intermediate environment between oxygen-deficient and oxygen-enriched, indicative of medium- to low-energy intertidal–supratidal deposition.

In addition, $\delta^{13}C$ has a correlation to sea level change. As shown in Figure 2, $\delta^{13}C$ turns from negative to positive in member Xi 3, keeps stable in member Xi 2, and decreases to be negative in member Xi 1. This variation illustrates that the sea level first rose and then fell. The Xiaoerblak Formation can be described as a complete third-order sequence, including a transgressive system tract (TST) equivalent to member Xi 1 and a highstand system tract (HST) equivalent to member Xi 2–member Xi 3.

In summary, the Early Cambrian Xiaoerblak Formation in the study area was deposited in a warm and humid near-shore shallow water environment, with high salinity and mild temperature. Specifically, the sedimentary environment was a restricted lagoon in member Xi 1, medium- to low-energy subtidal in submember Xi $2^1$–submember Xi $2^2$, medium- to high-energy subtidal in submember Xi $2^3$, and medium- to low-energy intertidal–supratidal in member Xi 3. According to the lithofacies sequence in the study area, the sedimentary sequence of the Xiaoerblak Formation is an inner-ramp lagoon–subtidal microbial mound shoal–tidal flat in a carbonated ramp setting (Figure 9).

### 5.2. Dolomitization Environment

Early dolomitization plays an importantly constructive role in the preservation of primary pores and pores formed in the early diagenetic stage, while late dolomitization has little contribution to and even destroys the preservation of early pores. Therefore, it is very important to determine the dolomitization time. The early dolomitization mainly occurred in the penecontemporaneous–shallow burial stage, and the late dolomitization mainly took place in the middle–deep burial stage. Their difference in diagenetic environment leads to distinctly different geochemical characteristics.

The $\delta^{13}C$, $\delta^{18}O$ and $^{87}Sr/^{86}Sr$ values of the Xiaoerblak Formation all fall within their value ranges of the Early Cambrian seawater. The low $\Sigma REE$ (<30 ppm) and wave-/hat-shaped REEs distribution patterns are similar to those of the Lower Cambrian micrite limestone, revealing seawater as the main dolomitization fluid. The measured high salinity Z (>125) and high contents of Sr (>70 ppm) and Na (>200 ppm) suggest a high seawater

salinity. The low order degree (<0.7) reflects that dolomitization was characterized by rapid replacement and rapid crystal growth at a low temperature. The ages of the five samples of host microbial dolomites determined by U-Pb dating are within the Cambrian age range, which may indirectly indicate that dolomitization occurred in the early diagenetic stage [37]. In addition, based on the fact that most of the samples are mainly micritic and silty dolomites with the original microbial and grain textures preserved, it can be judged that the Xiaoerblak Formation dolomite is the product of dolomitization in the penecontemporaneous–early diagenetic stage under a relatively low temperature, and the main dolomitization fluid was high salinity seawater.

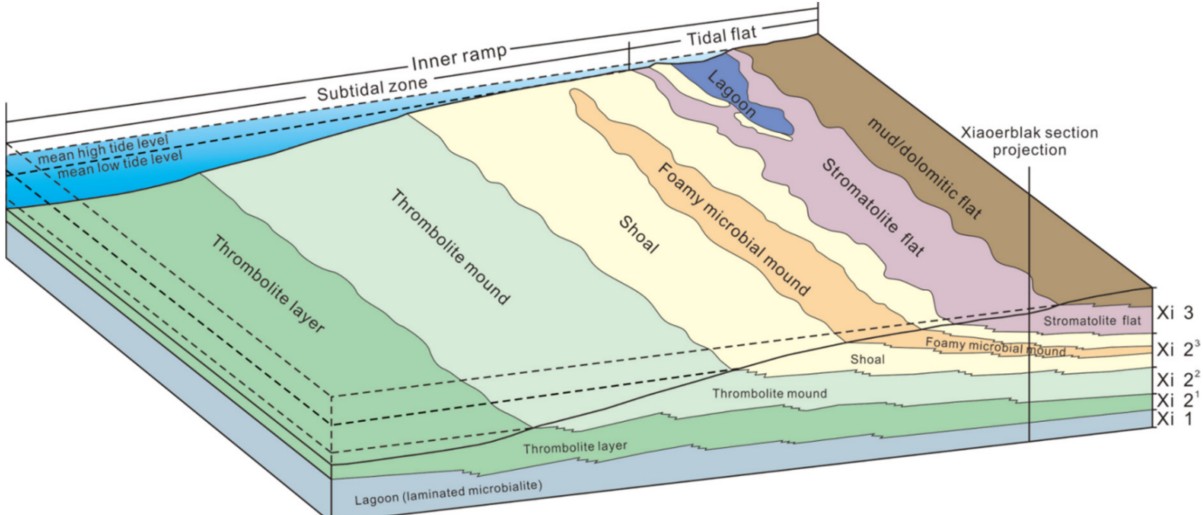

**Figure 9.** Sedimentary model of Xiaoerblak Formation in the study area. Xi 1, Xi 2 and Xi 3 are three members of Xiaoerblak Formation; Xi $2^1$, Xi $2^2$ and Xi $2^3$ are three submembers of member Xi 2.

*5.3. Pore-Forming Environment*

As mentioned above, the formation of pores in the Xiaoerblak Formation dolomite reservoirs is still controversial—primary pores in the deposition period, or secondary dissolved pores in the supergene period, or dissolved pores in the late burial stage—which directly affect the identification and distribution prediction of the main controlling factors of reservoirs. The carbonate cements in the host rocks and their pores, fractures and vugs were formed in different stages and diagenetic environments; therefore, the pore-forming stage and environment can be determined through a comparative analysis of their geochemical characteristics. Three types of cements were identified from the macroscopical characteristics of the Xiaoerblak Formation on the outcrop, and are mainly distributed in member Xi 1–Submember Xi $2^1$, including coarse-grained calcite in dissolved pores, coarse-grained calcite in fractures and saddle dolomite in dissolved pores/fractures, of which coarse-grained calcite in dissolved pores exhibits the highest proportion.

Based on the age comparison of host rocks and carbonate cements in microbial dolomite, it can be determined that the approximate formation time of vugs and fractures, as well as the formation time of the vugs, is between the age of the host rock and the age of the carbonate cement. The ages of three sparry calcite samples are younger than their host rocks; it can be inferred that the vugs were formed in Series 2 or Miaolingian of Cambrian. Therefore, it can be confirmed that the pores widely developed in the microbial dolomites of the Middle–Lower Xiaoerblak Formation were formed in the early diagenetic stage. The age of calcite cements in two fractures is close to the Late Triassic age, indicating a relation to the Indosinian tectonic movement. The age of saddle dolomite cements in two pores corresponds to the Paleogene age, revealing a relation to the tectonic movements during the late Himalayan period, when the hydrothermal dolomitization played a dominant and destructive role on reservoir porosity.

Based on the fact that the calcite cements in pores are characterized by low formation temperature (34 °C~45.3 °C, avg. 40.4 °C) and negative $\delta^{18}O_w$ in fluid ($-5.13‰$, SWOM) as measured by clumped isotope analysis, it can be confirmed that they were formed in a meteoric diagenetic environment according to the determination method of diagenetic environment using T and $\delta^{18}O_w$ proposed by Hudson [44]. The high temperature (74 °C) and relatively negative $\delta^{18}O_w$ ($-2.82‰$, SWOM) of the calcite cements in fractures suggest that they were formed in an atmospheric water buried environment. The saddle dolomite cements in pores/fractures have high temperatures (83.4 °C~85.3 °C, avg. 84.4 °C) and relatively positive $\delta^{18}O_w$ (avg. 1.63‰, SWOM), indicating that they were formed in a brine buried environment. The three types of carbonate cements were formed in three different diagenetic stages and environments, reflecting that their corresponding reservoir spaces were also formed in three different stages and environments.

Based on the petrological and geochemical characteristics of the Xiaoerblak Formation, the bedding-parallel dissolved pores/vugs widespread in the Middle–Lower Xiaoerblak Formation are the products of the penecontemporaneous stage, and are mainly microbial framework pores transformed by the atmospheric freshwater dissolution. This understanding further reveals that the lithofacies and high-frequency sequence boundary are important factors controlling the development of reservoirs in the Xiaoerblak Formation.

## 6. Conclusions

1.  The Xiaoerblak Formation mainly develops nine types of dolomites, i.e., laminated microbial dolomite, layered thrombolite dolomite, massive thrombolite dolomite (MTD), grain dolomite with bonding structure, crystalline dolomite remaining grain apparition, foamy microbial dolomite, stromatolite dolomite, grain dolomite with clay and argillaceous dolomite. According to the lithofacies association, it can be divided into three members: Xi 1, Xi 2, and Xi 3, of which member Xi 2 is subdivided into three submembers—the thickness is about 158.3 m;

2.  Analyses based on the petrological characteristics combined with the measured $\delta^{13}C$ and $\delta^{18}O$ values, contents of trace elements (e.g., Al, Fe, Pb, Cu, GA, Zn, Cr, and V), and content and distribution patterns of REEs in the Xiaoerblak Formation are effective tools for determining sedimentary environment. According to the measured data, the Xiaoerblak Formation in the study area is believed to have deposited in a near-shore shallow seawater environment with high salinity and temperature under a warm and humid climate. It divided into a third-order sequence, its sedimentary evolution sequence experienced an inner ramp lagoon, subtidal microbial mound shoal, and tidal flat from early to late, in a carbonate ramp setting;

3.  The petrological characteristics are combined with the measured order degree, $\delta^{13}C$ and $\delta^{18}O$ values, Sr and Na contents and REEs distribution patterns to determine the time and fluid of dolomitization. In the study area, dolomitization of the Xiaoerblak Formation occurred in the penecontemporaneous–early diagenetic stage, where high-salinity seawater acted as the main dolomitizaiton fluid;

4.  The genesis of porosity can be determined using U-Pb dating and clumped isotope temperature measurement techniques. Primary microbial framework pores/vugs formed by atmospheric water dissolution are the main reservoir spaces. The development of reservoirs is mainly controlled by lithofacies, a high-frequency sequence boundary and early dolomitization;

5.  The understanding of lithofacies types and sedimentary environment is of great significance to the research on sedimentary facies distribution in northwestern Tarim Basin. The understanding of dolomite reservoir genesis provides a basis for predicting the distribution of favorable reservoirs in the Xiaoerblak Formation and effectively supports the evaluation of exploration zones in the northern Tarim Basin.

**Author Contributions:** Conceptualization, J.Z. and Y.Z.; methodology, J.Z.; software, Y.Z.; validation, J.Z.; formal analysis, G.Y. and F.H; investigation, J.Z., Y.Z., L.H. and G.Y.; resources, G.Y. and F.H.; data curation, L.H.; writing—original draft preparation, J.Z.; writing—review and editing, J.Z. and Y.Z..; visualization, Y.Z. and L.H.; supervision, J.Z.; project administration, J.Z. and Y.Z.; funding acquisition, J.Z. and Y.Z. All authors have read and agreed to the published version of the manuscript.

**Funding:** This research was funded by the Scientific Research and Technology Development Project of PetroChina Company Limited (Grant No. 2021DJ0501).

**Data Availability Statement:** Data are available upon reasonable request. The data that support the findings of this study are available on request from the corresponding author. The data are not publicly available due to privacy or ethical restrictions.

**Acknowledgments:** We would thank Wenqing Pan and Anjiang Shen for their valuable suggestions, Anping Hu and Feng Liang for their guidance in the experiments, Zhangfen Qiao, Jin Li, Haoyuan Yu and Ran Xiong for their help in field work. We acknowledge this generous funding from PetroChina Company Limited.

**Conflicts of Interest:** The authors declare no conflict of interest.

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
