# Peer review of "Geochemical Characteristics and Their Geological Significance of Lower Cambrian Xiaoerblak Formation in Northwestern Tarim Basin, China"

_minerals, doi:10.3390/min12060781_

Round 1
Reviewer 1 Report
The article is important for the Lower Cambrian Xiaoerblak fromation in northwestern Tarim basin, China and appropriate to the journal theme (geochemistry and geochronology). The results are very important however, minor revisions are suggested in the attached manuscript. Also should be highlighted the article importance (mineral deposits, mineral exploration, management, development, regulation, etc.) in conclusions.

Reviewer 2 Report
The article is useful for carbonate depositional environments, and hence suitable to be published in Mineral.
Thank you for the effort you did.
